# Roles of Actin in the Morphogenesis of the Early *Caenorhabditis elegans* Embryo

**DOI:** 10.3390/ijms21103652

**Published:** 2020-05-21

**Authors:** Dureen Samandar Eweis, Julie Plastino

**Affiliations:** 1Laboratoire Physico-Chimie Curie, Institut Curie, PSL Research University, CNRS, 75005 Paris, France; dureen.samandar-eweis@curie.fr; 2Sorbonne Université, 75005 Paris, France

**Keywords:** actin cytoskeleton, myosin, *C. elegans* embryo, asymmetric cell division

## Abstract

The cell shape changes that ensure asymmetric cell divisions are crucial for correct development, as asymmetric divisions allow for the formation of different cell types and therefore different tissues. The first division of the *Caenorhabditis elegans* embryo has emerged as a powerful model for understanding asymmetric cell division. The dynamics of microtubules, polarity proteins, and the actin cytoskeleton are all key for this process. In this review, we highlight studies from the last five years revealing new insights about the role of actin dynamics in the first asymmetric cell division of the early *C. elegans* embryo. Recent results concerning the roles of actin and actin binding proteins in symmetry breaking, cortical flows, cortical integrity, and cleavage furrow formation are described.

## 1. Introduction

Actin is one of the most abundant proteins in the cell, existing as globular monomers (G-actin) that polymerize into helical filaments (F-actin). F-actin is polar with a fast-growing, dynamic ‘barbed end’ and a slow-growing, less dynamic ‘pointed end’. The dynamic assembly and disassembly of F-actin, as well as the myosin molecular motors that associate with actin, produce forces within the cell and between cells that drive cellular and tissue reorganization. Actin dynamics are controlled by actin-binding proteins, which variously activate or inhibit F-actin formation, stabilize/destabilize existing actin structures, or bind actin monomers [1]. One of the most important actin regulators is the Arp2/3 complex that nucleates the formation of new actin filaments as branches off the sides of existing filaments. Another important class of actin polymerization nucleators is the formin family of proteins. Formins create new unbranched filaments, and also associate with the barbed end of the actin filament, enhancing actin assembly. Once formed, F-actin is remodeled by actin bundling and cross-linking proteins—such as fascin, plastin/fimbrin, and filamin—which promote the formation of parallel and antiparallel bundles of F-actin or cross-linked arrays, respectively. Additionally, myosin remodels F-actin structures, using actin filaments as tracks, sliding antiparallel filaments in relation to each other to create contraction. G-actin binding proteins include profilin, which is key for actin dynamics in the cell, as it reduces spontaneous nucleation and also prevents pointed end polymerization thus allowing for controlled, directed actin assembly in vivo. Capping proteins, which bind barbed ends and prevent further polymerization, and ADF/cofilin proteins, which sever actin filaments, are also important for in vivo actin dynamics and function [2]. Many actin binding proteins, including the main polymerization nucleators and myosin, are regulated by the small GTPases Rho, Rac, and Cdc42, which are in turn controlled by guanine-nucleotide-exchange factors (GEFs) and GTPase-activating proteins (GAPs) downstream of extracellular signals.

The diversity of actin-binding proteins leads to a diversity of actin architectures in the cell, adapted to different functions [1]. The actomyosin cortex is a thin layer of cross-linked actin filaments interspersed by myosin, attached to the inner face of the cell membrane. In the moving cell, the cortex at the back of the cell contracts to squeeze the cell forward, while protrusive actin structures, lamellipodia and filopodia, form at the front. Blebs, another type of cell protrusion, occur when the cell membrane detaches from the cytoskeleton and balloons outward, initially devoid of actin, due to actomyosin contractility in the cortex.

*Caenorhabditis elegans* has been used as a model organism to investigate the regulation and dynamics of actin networks in developmental processes [3]. In particular, the first asymmetric division of the *C. elegans* single cell embryo has been studied extensively in order to understand symmetry breaking, polarity establishment, microtubule assembly, spindle positioning, and the cell shape changes that accompany asymmetric cell division [4,5]. Briefly, as concerns the actin cytoskeleton and related proteins, in the just-fertilized zygote, cortical ruffles are evident all around the circumference of the embryo due to the highly dynamic and contractile cortical actomyosin layer. Symmetry is broken when the sperm contents approach the future posterior pole, locally downregulating contractility there and initiating the retraction of the actomyosin cortex to the future anterior pole [6,7]. This flow of actomyosin density towards the anterior pole leads to an invagination at the boundary between high and low actomyosin activity similar to ruffles but much deeper, called the pseudocleavage furrow. Anterior-directed cortical flow is concomitant with the segregation of the polarity proteins PAR-3, PAR-6, and PKC-3 to the anterior of the embryo, while PAR-1 and PAR-2 are recruited to the posterior cortex [8]. 

The result of this polarization phase in the embryo is the formation of two cortical domains that have different actomyosin activity and different PAR protein occupancy. During this period, known as the maintenance phase, a complex network of reciprocally supportive and antagonistic interactions between PAR proteins exists reinforcing their localization at the poles of the embryo. This also includes effects on actomyosin wherein PAR-1 and PAR-2 at the posterior pole have a role in inhibiting the posterior localization of non-muscle myosin II (NMY-2), while anterior PARs—PAR-3 and PAR-6/PKC-3—lead to the accumulation of NMY-2 at the anterior, establishing and maintaining a more contractile anterior pole [9]. During the maintenance phase, in a series of steps that are largely microtubule dependent, the maternal pronucleus joins the paternal pronucleus at the future posterior side of the cell, the complex recenters and then forms the spindle, which is pulled posteriorly again during anaphase to give the final asymmetry in division. Importantly, the positioning of the cleavage furrow and site of division is highly dependent on the positioning of the spindle [10], and it is the cortical polarity of the embryo that controls the mitotic spindle shift since cortical pulling forces are more pronounced at the posterior pole [6,11].

Although the first cell division of the *C. elegans* embryo has been studied for decades, many open questions remain. Here we review results from the last five years that address some of the outstanding questions in the field and demonstrate the multitude of roles played by the actin cytoskeleton in the *C. elegans* embryo. 

## 2. Role of Actin in the Just-Fertilized Embryo during Completion of Meiosis

As for many organisms, *C. elegans* oocytes complete meiosis upon fertilization. While meiotic divisions are occurring at what will become the anterior pole of the embryo, the sperm contents, including genetic material, are retained at the site of sperm entry at the posterior pole, despite cytoplasmic streaming in the embryo.

A recent finding reported that the actin cytoskeleton was what was confining sperm DNA to the posterior pole of the embryo, and keeping it from getting captured by the meiotic spindle [12]. In embryos where actin polymerization was reduced, either by interfering with the profilin/formin mode of actin assembly or via application of inhibitory drugs, sperm DNA was distributed throughout the embryo due to cytoplasmic flows. This study further showed that the sperm contents were not restricted to the posterior cortex by cytoplasmic actin via a sieving effect as could have been expected. Rather, the study suggested that it was the cortical actin pool that was important for sperm content confinement by an as-yet-unidentified mechanism. Although most studies on the one-cell embryo pay particular attention to actin structures and functions during and after polarity establishment, this work showed that F-actin was an important player prior to these events. 

## 3. Actomyosin Dynamics in Symmetry Breaking

The morphological and biochemical changes defining the anteroposterior axis of the embryo occur downstream of sperm entry, which has been shown to break the symmetry of the embryo and define its future posterior pole [6,7,13]. At the moment of fertilization, actomyosin foci are present over the entire embryo surface, along with actomyosin-based cortical contractions. At the end of meiosis II and just as mitosis is beginning, the sperm centrosome moves close to the embryo cortex at the future posterior pole. As described in the introduction, this produces an immediate cessation of cortical activity and a flow of actomyosin foci away from this region. As reviewed recently [6], this local transformation in the actomyosin cytoskeleton is known to be due to the removal of the RhoGEF ECT-2 from the cortex. 

Until recently, the molecular nature of the cue delivered by the sperm centrosome to downregulate posterior actomyosin activity and initiate cortical flow was unknown. Novel studies in the past year have identified the mitotic kinase Aurora A (AIR-1) as the previously unidentified centrosome component (Figure 1). One study showed that the phosphorylated, active form of AIR-1 was released from the centrosomes into the cytoplasm, driving the inhibition of posterior cortical actomyosin networks in the vicinity of the centrosomes [14]. This was discovered based on the finding that the GFP-tagged version of AIR-1 was not completely wild-type. GFP-labeled AIR-1 embryos performed AIR-1-dependent processes normally, including centrosome maturation, but failed to correctly clear actomyosin from the future posterior pole during symmetry breaking. It was hypothesized that this effect was due to a defect in diffusion of the GFP-labeled protein. This hypothesis was supported by experiments that manipulated the position of the centrosome, moving it closer and further away from the cortex, improving and exacerbating, respectively, the actomyosin clearing defect [14]. Another recent study came to a similar conclusion concerning the identity of the centrosome-derived cue [15]. Both studies showed that AIR-1′s role in symmetry breaking was a result of its effect on ECT-2, altering its localization and perhaps its GEF activity by an unknown mechanism. AIR-1 could also be downregulating myosin activity via its phosphorylation of other RHO-1 pathway effectors that are upstream of ECT-2.

In addition to its centrosomal role in initiating cortical flow, non-centrosomal AIR-1 has been shown in recent studies to globally downregulate cortical actomyosin activity during polarity establishment. Embryos lacking AIR-1 showed hypercontractility of the cortex, and became bipolar, with NMY-2-poor/PAR-2-rich domains at both poles of the embryo, and weak cortical flows directed toward the embryo center from both sides [14,15,16,17]. One study further showed that this bipolarization could occur even in wild-type embryos when fertilized with acentrosomal sperm [16]. Putting all this together, it seemed that there was a basal activity of non-centrosomal AIR-1, which kept actomyosin downregulated globally and prevented spontaneous bipolarization events, while centrosomal AIR-1 downregulated actomyosin locally to initiate the polarizing cortical flows that establish embryo polarity. Why PAR-2 domains form in a bipolar manner was unclear, but it was proposed that curvature could be the determining factor [16]. This hypothesis was tested by placing *air-1* depleted embryos in triangular chambers. It was observed that PAR-2 domains emerged in regions with the highest curvature. PAR-2 accumulation at curved regions could be biochemically driven by lipid affinities or due to geometrical considerations, where the curved surface of the poles restricts diffusion out of the immediate vicinity [16]. An additional function for AIR-1 was recently observed in the steps leading up to fertilization and symmetry breaking in the oocyte, where AIR-1 was shown to play a role via another cell cycle kinase PLK-1 (polo-like kinase) in regulating anterior PAR loading and activation at the membrane, although the details of the actomyosin cortex are not addressed in this study [17]. This regulation prevents premature polarization of the embryo, and enforces the dependence on the centrosome cue.

## 4. Cortical Flows during Polarity Establishment

Anterior-directed actomyosin contractility creates cortical flow, which sets up the polarity axis of the embryo as concerns both actin cytoskeleton and PAR proteins. Several recent studies have shed light on the precise details of how actomyosin-driven dynamics are coupled to PAR protein polarization. One study showed that cortical actomyosin tension facilitates the clustering of PAR-3 perhaps by inducing conformational changes that allow oligomerization [18]. Indeed, embryos lacking cortical tension molecules like NMY-2 did not exhibit PAR-3 clusters, but clusters could be rescued by artificial increases in cortical tension applied via osmotic shock for example. PAR-3 clustering induced clustering of PKC-3, and the clustering of both proteins was important for their proper transport to the anterior pole. A coincident study concurred, proposing that either clustering reduced the effective diffusion of membrane-associated anterior PARs, or the larger size of clusters allowed them to interact more effectively with the cortical actomyosin layer, both of which would favor advective transport by flows [19]. Another study came to a similar conclusion, showing via a single-cell extract technique that PAR-3 clusters were more efficiently transported by cortical flow due to the longer residence time of the PAR-3 oligomers at the cortex [20]. These and other findings concerning the roles of PLK-1 and CDC-42 in PAR protein clustering and cortical transport to the anterior pole are nicely reviewed in [21]. 

Another recent study showed the very clear link between cortical flows and PAR domain location. In this work, they used a novel focused-light-induced cytoplasmic streaming (FLUCS) system to induce controllable cytoplasmic flows in the embryo via temperature changes [22]. Cytoplasmic flows were shown to drive cortical flows, which were exactly mirrored by PAR protein domain relocation [22]. Compellingly, when the PAR-2 domain was moved to the anterior pole by flow, the embryo divided with an inverted size asymmetry (smaller anterior cell). All together, recent work confirms and extends the importance of actomyosin cortical flow for PAR domain establishment in the embryo. 

The studies discussed above were principally about how flows driven by actomyosin contractility play a role in PAR protein localization, however the converse is also true: PAR proteins affect NMY-2 recruitment to the cortex and thus control flows. This complex interplay was investigated in a recent study where the kinetics of NMY-2 association and dissociation from the posterior and anterior cortex were measured [23]. It was observed that NMY-2 association with the cortex was identical for the posterior and anterior domains. Dissociation on the other hand was twice as high in the posterior domain as compared to the anterior region. This was shown to depend on PAR-6 specifically, where increases in PAR-6 were accompanied by decreases in the dissociation of NMY-2. It was proposed that this mechanochemical feedback between actomyosin localization and PAR proteins ensured robustness of embryo polarity. 

A previously unidentified role for cortical flow was highlighted in a recent study demonstrating that flow contributed to centrosome separation [24]. The centrosome pair present on the paternal pronucleus must be separated to correctly form the bipolar mitotic spindle necessary for the first cell division. The authors showed that cortical dynein, imbedded in the actomyosin cortex, and also bound to microtubules emanating from the centrosomes, was the main player in this process. Cortical dynein was swept anteriorly by actomyosin flows, pulling with it the centrosomes. When cortical flow was impaired by depletion of NMY-2 or RHO-1, centrosome separation was retarded although there was no effect on cell cycle progression. Since AIR-1 on the centrosome is what is responsible for breaking the symmetry of the embryo and triggering cortical flow, centrosomes are perfectly positioned to harness flow for separation (Figure 1). 

PAR proteins control flows, but actin-binding proteins are also known to affect flows, presumably by modifying the organization and mechanical properties of the cortex. This point was addressed by carefully characterizing flow velocities, flow pulsatility and myosin foci size and density in *C. elegans* embryos upon depletion of different actin-binding proteins and myosin regulators [25]. Many were found to affect the measured parameters in sometimes subtle ways. One general result that came out of this analysis was that bigger, more sparsely-distributed myosin foci were correlated with slower average flow velocities. However, the main point of this study was that data clustering revealed classes of proteins with sometimes dissimilar molecular activities that nevertheless affected cortical properties in similar ways. This pointed to a degeneracy in the molecular components needed to produce a given phenotype. A related paper specifically examined the origins of pulsatility, asking why this did not lead to instability and collapse of the system [26]. They showed evidence that Rho activity oscillated, thus damping down myosin foci contraction and preventing collapse of the cortex. Although not pertaining to the one-cell embryo, another study came to a similar conclusion about the presence of a Rho oscillator [27]. These studies all together illustrate that actomyosin contractility and resulting flows are robustly controlled in the *C. elegans* embryo. 

## 5. Cortical Actin Architecture and Dynamics

Many proteins that affect flows in the polarizing embryo also have effects on the stability of the actomyosin cortex in the maintenance phase, for example, the actin filament bundling protein plastin (PLST-1) [28]. Deletion of *plst-1* led to smaller and more dispersed NMY-2 foci as well as weaker and less coherent directed cortical flows suggesting a need for PLST-1 for proper NMY-2 coalescence and resulting flows. Predictably, since PAR protein organization depends in part on cortical flows, there was also a defect in the polarity of PAR proteins in embryos lacking PLST-1. Via laser ablations and measurements of cortex recoil velocity, lack of PLST-1 was shown to decrease the tension of the anterior cortex of the embryo during the maintenance phase, indicating reduced cortical stiffness. Overall, the authors proposed that, by controlling actin network connectivity, PLST-1 regulated the mechanical properties of the cortex. Imaging techniques that allow visualization of nanoscale structures of the actin cortex in the presence and absence of PLST-1 would be very interesting to obtain in order to get a molecular picture of connectivity differences.

Another interesting study on cortical stability addressed the functions of cadherin (HMR-1) in the *C. elegans* embryo [29]. HMR-1 is normally involved in cell-cell adhesions, but these are not present in the one-cell embryo. It was further shown that HMR-1′s ability to interact with the permeability barrier surrounding the plasma membrane of the embryo was not important for HMR-1′s role at this stage, so it appeared that HMR-1 was playing an adhesion-independent role. The observation was that clusters of HMR-1 formed during the polarity establishment phase of the single cell embryo, and associated with cortical F-actin. Clusters were transported to the anterior pole by flows where, during the maintenance phase, they appeared to antagonize cortical NMY-2. Indeed, upon depletion of *hmr-1*, the level of anterior cortical NMY-2 significantly increased while there was no significant effect on the global concentration of NMY-2 or cortical F-actin. Non-junctional HMR-1 clusters appeared to control cortical NMY-2 via its upstream regulator, RHO-1: the amount of active cortical RHO-1 increased upon *hmr-1* depletion. Finally, in conditions of depleted *hmr-1*, cortical flows were accentuated and the actin cortex was observed to tear away from the cell membrane, probably due to the combined effect of increased flow/contractility and loss of stabilizing linkages conferred by HMR-1, which bridges the cell membrane to the actin network. Overall, this study brought to light a role for non-adhesive cadherin clusters in regulating actomyosin cortex stability and flows, and attachment to the cell membrane.

While much of the work reviewed here deals with the actomyosin cortex, which is attached to the cell membrane, not many studies have investigated the involvement of the cell membrane itself in the *C. elegans* embryo. A new study showed that the lipid phosphatidylinositol 4,5-bisphosphate (PIP2) was non-uniform in the polarized embryo. It was enriched at the anterior pole where it controlled actin dynamics and PAR protein recruitment [30]. However, this study conflicts with another recent work, which demonstrated that the increased formation of membrane folds at the anterior pole lead to an enrichment of lipids in general, not just PIP2 [31]. In fact, these membrane folds were shown to be filopodia, dependent on the formin CYK-1 and the Arp2/3 complex. The existence of PIP2 membrane microdomains in the *C. elegans* zygote therefore does not appear likely.

Overall, these recent studies detail the molecular regulation of the structure and dynamics of the actomyosin cortex during the post-polarization phase, expanding what is already known for symmetry breaking and polarity establishment. 

## 6. Contractile Ring Formation and Positioning

Once polarity is triggered and established, the main steps preceding cytokinesis are contractile ring formation and spindle positioning, which is what defines the position of the cleavage furrow [10]. The ring is comprised of F-actin, myosin and accessory proteins and it accomplishes cytokinesis of the single cell embryo. Detailed, up-to-date reviews of the players needed for ensuring proper formation of the cytokinetic ring and furrow ingression have been published [6,32]. 

Insight as to how the contractile ring is constructed was provided by a recent study, where it was shown that F-actin alignment due to converging cortical flow at the cell equator was sufficient to drive ring formation (Figure 2) [33]. No localized actin polymerization was necessary, and in fact did not exist for ring formation during pseudocleavage, although such mechanisms exist for contractile ring formation during cleavage. Reducing flow rates by perturbing myosin machinery had predictable effects on the pseudocleavage furrow: slight reductions still permitted actin filaments to align to create a furrow while drastic reductions abolished furrow formation. A related work showed that mechanical compression of embryos inhibited longitudinal flows, but favoured rotational flows, which had the same end result of efficiently aligning filaments to create a contractile ring [34]. This study further showed that myosin flow characteristics were not quite the same as actin flows. In particular, NMY-2 flows were more long-ranged, a difference explained by the fact that actin was disassembled by compression as it flowed to the equator while myosin was not [34]. Cortical flows were also shown to be important for ring dynamics during constriction [35]. This study found that new cortex, including NMY-2, was being pulled into the ring due to myosin activity and polar relaxation. The added myosin in turn led to increased flow into the ring, and an exponential increase in the amount of ring components. This was counterbalanced by disassembly-coupled ring shortening. The end result was that, although NMY-2 levels and the levels of other ring components appeared to remain constant during ring constriction, the ring was in fact undergoing dramatic restructuring. These recent studies emphasize the robustness of cortical flows for creating and maintaining a proper furrow structure.

One recent study revealed that both the Arp2/3 complex and the formin CYK-1 were important for the kinetics of cytokinesis even though the Arp2/3 complex did not localize to the cleavage furrow [36]. CYK-1 was shown to be enriched in the contractile ring and essential to elongate filaments to form it: *cyk-1* depletion led to a decrease in actin bundles at the equatorial cortex, and a furrow that initiated and ingressed more slowly. However, the Arp2/3 complex was also found to positively regulate the kinetics of contractile ring assembly and ring constriction, although it had the opposite effect on furrow initiation. Intriguingly, the slowing down of cytokinesis upon Arp2/3 complex inhibition was a result of an increase in CYK-1-mediated actin polymerization at the cell cortex and in the contractile ring. The conclusion was that Arp2/3 complex activity was required in the cortex to temper CYK-1-based polymerization, possibly via a competition for actin monomers as has been shown in other systems [37]. Too much cortical actin could be supposed to interfere with the cell shape changes accompanying cell division, while excessive actin in the furrow could inhibit contraction by impeding ring disassembly.

Proteins that affect cortical flows in the polarization phase and actomyosin cortex integrity in the maintenance phase also affect the speed of contractile ring formation and its contraction, due to the dependence of all these processes on similar parameters. The new studies concerning PLST-1 and HMR-1, mentioned previously in the context of cortex integrity, showed that these proteins also play roles in furrow formation. Lack of PLST-1 resulted in furrows that formed more slowly, although once formed, they ingressed at the normal speed, while lack of HMR-1 produced faster ingression than wild-type although with a tendency for the cortex to peel away from the cell membrane at the anterior pole [28,29]. 

As concerns furrow ingression, some studies have shown evidence that it is not myosin motor activity that drives contraction of the cytokinetic ring, but rather myosin’s actin cross-linking activity coupled with depolymerization [38,39,40]. New studies in the *C. elegans* embryo have demonstrated unequivocally that NMY-2 motor activity was required throughout the formation and constriction of the actomyosin ring [41]. During ring assembly, the activity of NMY-2, rather than its F-actin cross-linking ability, were shown to control proper densification and alignment of actin in the cytokinetic ring, and timely deformation of the cell equator. In general, however, actin cross-linking activity is known to play a role in modulating contractility, and it was shown that for the *C. elegans* cytokinetic ring, an optimal degree of cross-linking existed, while too much or too little cross-linking activity was deleterious to contraction [42].

Contractile ring formation is important, but so is its correct positioning in the cell. Although positioning is mostly determined by astral microtubule dynamics, two recent papers show the importance of the downregulation of actomyosin contractility at the embryo cortex for correct placement of the furrow. One study brought to light the essential role of AIR-1 in clearing contractile ring proteins such as anillin, but also actin, from the poles of the embryo [43]. At the onset of anaphase, a conserved activator of AIR-1, TPXL-1, became localized to astral microtubules, and it was shown that its capacity to bind and activate AIR-1 was key for polar clearing. The hypothesis was that activated AIR-1 promoted clearing of contractile ring proteins from the poles of the embryo via phosphorylation of target proteins, but this mechanism has yet to be investigated in detail. Taken all together with results described previously in this review, this study provides another example of the AIR-1′s varied roles in the regulation of the actin cytoskeleton during the development of the early *C. elegans* embryo. 

The role of cortical dynamics in furrow positioning was also demonstrated in another recent study, which described a role for bleb formation in releasing cortical tension in mutant embryos that had excessive anterior myosin contractility [44]. In these embryos, the furrow initially formed in an anterior position, however DNA segregation defects were avoided by a posterior shift of the furrow concomitant with an anterior shift of the anterior nucleus. These shifts were shown to be produced by bleb formation near the anterior side of the furrow. The release of tension due to bleb formation allowed repositioning of the nascent cleavage furrow, and also created cytoplasmic flows that contributed to moving the nucleus posteriorly.

## 7. Conclusions

The *C. elegans* embryo is a powerful model system for understanding the many roles of the actomyosin cytoskeleton in asymmetric cell division. Results obtained using this model over the last five years have answered outstanding questions, as summarized in this review. In particular, the nature of the centrosomal cue for symmetry breaking has at last been identified. Moreover, other recent studies have delved deeper into the molecular mechanisms of how actomyosin dynamics drives morphological changes in the embryo. In the past, the microtubule cytoskeleton and PAR proteins have taken center stage in the *C. elegans* embryo. Our summary in this review shows the importance of the actin cytoskeleton in polarity establishment of the *C. elegans* zygote as well as the cellular shape changes that ensure the first asymmetric division of the embryo. 

## Figures and Tables

**Figure 1 ijms-21-03652-f001:**
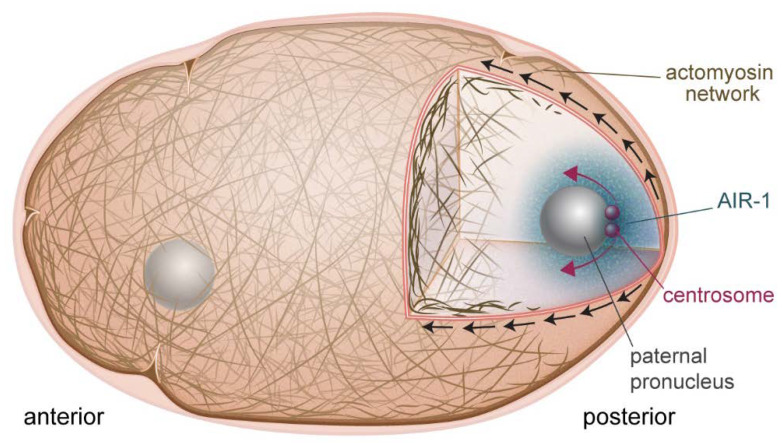
Symmetry breaking in the one-cell embryo. AIR-1 (blue cloud) is the cue that initiates polarization of the embryo. AIR-1 diffuses from the centrosome (red spheres) and downregulates actomyosin at the adjacent cortex. This causes a local weakening, and produces cortical flows (black arrows) directed away from this point, which also serve to separate the centrosomes (red arrows).

**Figure 2 ijms-21-03652-f002:**
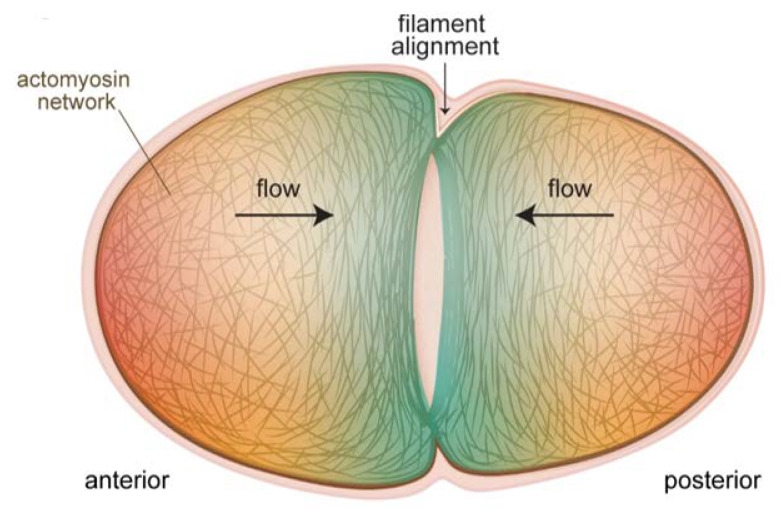
Cortical flows align filaments to form the contractile ring for cytokinesis of the one-cell embryo. The actomyosin cortex at the poles flows toward the equator of the embryo (black arrows), and these flows progressively transform the unorganized actin filament network at the poles (red shading) into the aligned structure of the cytokinetic ring (blue shading). In cases where actin-binding proteins are perturbed, thus altering cortical properties and flows, the formation and ingression of the contractile ring are impacted (see text).

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
