# Peer review of "Roles of Actin in the Morphogenesis of the Early Caenorhabditis elegans Embryo"

_ijms, 2020, doi:10.3390/ijms21103652_

Round 1

Reviewer 1 Report

The review of  morphogenesis of C. elegance embryo, not a new idea and/or new data, but the topic seems to be interesting to readers.

1. I think the proposed topic is not fully addressed. For example, challenging “actin and c. elegance and embryo” in Pubmed provide about 200 articles, and the review is not restricted to these embryos. I feel that 42 citation is low for a review in general and certainly on this highly studied topic. Check other references that describe the morphogenesis of C. elegance.

2. The abstract should provide a short description of the manuscript. In my opinion, there is a lack of information about the association of actin in the morphogenesis in this part. Instead of this, the reader can read the activity of actomyosin systems in the morphogenesis of C. elegant embryo. The abstract should be improved.

3. Schematic illustration or photographs that show the typical activity of actomyosin system should be included in the following section. Text only description is unclear that imagines such activities.

       3. Actomyosin dynamics in symmetry breaking 

       4. Cortical flows during polarity establishment 

       6. Contractile ring formation and positioning

The above sections need schematic illustrations and/or photographs that show actin activity during embryogenesis. 

Reviewer 2 Report

This review comprehensively summarizes the recent studies (published during the last 5 years) that have completed our understanding of the role of actin and myosin in the context of the C. elegans one-cell asymmetric division. This manuscript gives a nice overview of the main concepts that emerge from those studies and is clearly suitable for publication in the “International Journal of Molecular Sciences”. Below are however a few comments that should be addressed before publication to clarify and complete some sections of the manuscript.

Lines 88-92: This paragraph is not completely accurate as the paper from Panzica et al., shows that the actin cytoskeleton maintains the sperm DNA in the posterior region of the embryo but is not required to keep the DNA close to the cortex per se. Indeed, interfering with the actin cytoskeleton results in the movement of the sperm DNA towards the anterior of the embryo but does not change the distance between the DNA and the cortex. This part should be rephrased to precisely describe this data.

Lines 109-119: The authors should explicitly mention that the work from Zhao et al.; suggests that the active form of AIR-1 is released from the centrosomes and then inhibits actomyosin contractility at the nearby (=posterior) cortex. This would provide the rationale behind the two sentences mentioning 1) the possibility that reduced diffusion of GFP::AIR-1 explains its inability to properly regulate actomyosin and 2) the effect of centrosome displacement.

Lines 124-134: This paragraph is somewhat misleading as it gives the impression that the global cortical hypercontractility is responsible for the bipolarity observed in air-1 embryos, which is, to my understanding, not the case. The authors could make it clearer by:

  • More explicitly describing/comparing the effect of centrosome associated AIR-1 on the posterior cortex (local dissociation of actomyosin before polarization, this induces cortical flow) and the effect of non-centrosomal AIR-1 on the whole cortex (global actomyosin dowregulation + dissociation of actomyosin foci during late prophase (Zhao et )).
  • Mentioning that loss of AIR-1 is also associated with loss of polarizing cortical flows and appearance of weak flows emanating from the two poles and directed towards the center of the embryo (see Kapoor and Kotak, Klinkert et al.,). Note that loss of polarizing cortical flows is also observed when centrosomal proteins are depleted.
  • Making it clear that the bipolarity observed in the absence of AIR-1 is mostly due to the absence of centrosomal AIR-1. Indeed, centrosome ablations and, as the authors mention, fertilization with acentrosomal sperm also lead to bipolarity (Klinkert et al.,). They could also add that PAR-2 localization in air-1 embryos is independent of the centrosome position (Kapoor and Kotak, Klinkert et al.,)
  • Concluding that AIR-1 and centrosomes are required for cortical flow and correct polarization of the embryo.
  • Discussing the mechanism by which PAR-2 accumulates at the two poles of the embryo in the absence of AIR-1 or centrosomes (work from Klinkert et al.,). Note however that 1) the term “symmetry breaking” used in Klinkert et al to describe the bipolar accumulation of PAR-2 is somewhat misleading (the embryo is still symmetric…). 2) the absence of cortical flow is unlikely to be sufficient to explain this bipolarity (direct flow inhibition prevents polarization of the embryo but is not sufficient to observe bipolar accumulation of PAR-2).

Line 134-138: Slightly rephrasing the sentence may help emphasizing that this describes an additional function of AIR-1.

Lines 149-150: What do the authors mean by “clustering was integral to proper transport”?

Lines 191-196: The authors could also mention the effect of actin-binding proteins on actomyosin foci pulsatility described in Naganathan et al.

Lines 198-201: In addition to the work of Nishikawa et al;, the authors could also mention the work of Michaux et al. (The Journal of Cell Biology, 2018). Although this work has been performed in 2-cell embryos, it also describes the role of Rho dynamics on actomyosin pulsatility.

Lines 198-201: Wouldn’t it be more appropriate to discuss actomyosin pulsatility in the next section (section 5 about cortical actin dynamics)?

Lines 224-230: The authors could mention that HMR-1 clusters also impede the mobility of cortical actomyosin.

Line 227: “hmr-2” should be “hmr-1

Lines 252-261: This paragraph could be completed by a comparison of actin and myosin dynamics during the induction of ring formation by cortical flows (see Singh et al.).

Lines 272-274: This sentence should be rephrased to make it clear that (and how) ring kinetics is also affected upon CYK-1 depletion. Also note that furrow initiation is faster in ARP-2 depleted embryos (and not slower as written here).

Lines 285-286: The description of the phenotypes caused by the loss of PLST-1 does not seem to be completely accurate. Indeed, loss of PLST-1 either leads to defective furrow ingression (no ingression at all or regression, but as far as I can see, the speed of ingression has not been measured in those embryos) or to a delay in furrow initiation (but those embryos do not show a difference in the speed of ingression).

Lines 288-294: This paragraph could be completed by mentioning the work from Descovich et al (Mol Biol Cell, 2018) about the role of cross-linkers during cytokinetic ring contriction.
